# WEFix: Intelligent Automatic Generation of Explicit Waits for Efficient Web End-to-End Flaky Tests

## ABSTRACT

Web end-to-end (e2e) testing evaluates the workflow of a web application from beginning to end. It simulates real-world user scenarios to ensure the application flows behave as expected. Web e2e testing plays an indispensable role in the development of modern web applications. However, web e2e tests are notorious for being flaky, *i.e.,* the tests can produce inconsistent results, passing or failing unpredictably, despite no changes to the code under test. One common type of flakiness in web e2e testing is caused by nondeterministic execution orders between the test code and the client-side code under test. In particular, UI-based flakiness emerges as a notably prevalent and challenging issue to mitigate. Such flaky tests are challenging to fix because the test code has limited knowledge about the client-side code execution.

In this paper, we propose WEFix, a technique that can automatically generate fix code for UI-based flakiness in web e2e testing. The core of our approach is to leverage browser UI changes to predict the client-side code execution and generate proper wait oracles. We evaluate the effectiveness and efficiency of WEFix against 122 web e2e flaky tests from seven popular real-world projects. Our results show that (1) UI-based flakiness is prevalent; (2) implicit waits introduce significant runtime overhead; (3) WEFix dramatically reduces the overhead (from 3.7× to 1.25×) while still achieving a high correctness (98%).

## 1 INTRODUCTION

Web end-to-end (e2e) testing [52] involves testing a web application's workflow from beginning to end. Web e2e testing evaluates if an application works as expected by simulating different user behaviors. As the complexity of web applications grows, web e2e testing is becoming increasingly important in web development [13, 32]. For example, Facebook mandated the full deployment of web e2e testing in continuous delivery as early as 2015 [17].

Despite their importance, web e2e tests are notorious for being flaky. Flaky tests are software tests that produce inconsistent results, passing or failing unpredictably, despite no changes to the code under test. For example, at Google, it was reported that around 16% of their tests were flaky [34]. According to a recent study, a widely-used network application produced as many as 18 failures out of 100 executions of the test suite [38].

One common type of flakiness in web e2e testing is caused by nondeterministic execution orders between the server-side test code and the client-side browser code under test [2, 19, 28]. In particular, UI-based flakiness emerges as a notably prevalent and challenging issue to mitigate, which has been widely acknowledged by both researchers [41] and developers [26, 48]. UI tests refer to tests that validate the correct behavior of a user interface, while UI-based flaky tests are UI tests that are flaky. In this paper, our research pivots towards UI-based flakiness.

In fact, UI-based flakiness has already persisted for over a decade. In 2009, the Google testing team proposed to resolve such flaky tests by adding wait statements, *e.g.,* adding a 2-second delay, to wait for the completion of each browser event (such as AJAX requests and page load) [23]. We define the approach of waiting for a fixed amount of time as *implicit waits*, also known as timeouts or thread sleeps. While implicit waits appear to be a straightforward remedy to flakiness, our experiment shows that implicit waits can introduce up to 36× runtime overhead (Table 3), making it impractical for real-world web testing environments.

As a result, intentional wait should be added only at locations where flakiness is likely to occur. We define this more efficient approach as *explicit waits*, which wait until a certain oracle or condition is met, thereby accommodating dynamic environmental factors, *e.g.,* network delays and server loads. In practice, however, manually selecting appropriate explicit waits creates significant challenges for developers: First, it is difficult to pinpoint where the flakiness may occur and what explicit conditions should be expected. Second, manual insertion of explicit waits can be error-prone because extensive use of explicit waits are needed to combat the ubiquity of such flakiness.

To address these challenges, we propose WEFix, which automates the insertion of appropriate explicit waits at flaky-prone locations, achieving a balance of high correctness and low overhead. Specifically, WEFix first records the browser-side DOM mutations triggered by each command on the fly. These DOM mutations will be used by the server-side test code to predict the client-side operations. To record the mutations, WEFix uses cookies as the communication channel between the test code and the browser-side code under test. WEFix then generates wait oracles for the test code based on these mutation records. Finally, WEFix employs a finite state machine to model the DOM mutation events, ensuring that no additional mutations occur once the oracle is met and that each command waits for all GUI changes to conclude before proceeding to the next command. Our evaluation on seven popular GitHub web projects shows that 65.7% (1,145 out of 1,743) of the commands are flaky-prone, indicating UI-based flakiness is common. We reproduce 122 UI-based flaky tests to evaluate the effectiveness and efficiency of WEFix. Our results show that WEFix successfully fixes 98% (120 out of 122) of the flaky tests and reduces the overhead from 3.7× to 1.25× compared to implicit wait approaches.

In summary, this paper makes the following contributions:

- We propose WEFix, which automatically inserts explicit waits to fix flakiness by generating wait oracles based on DOM mutations.
- We evaluate WEFix on seven popular GitHub projects, and the results show that (1) UI-based flakiness is prevalent; (2) implicit waits introduce significant runtime overhead; (3) WEFix dramatically reduces the overhead (from 3.7× to 1.25×) while still achieving a high correctness (98%).

- We make WEFix publicly available on NPM [7], which features a user-friendly UI panel to help developers analyze UI-based flaky tests.

## 2 BACKGROUND

### 2.1 Web E2e UI Test

A typical web e2e UI test consists of a sequence of *commands* followed by *assertions*. Commands are driver functions used to emulate web user behaviors on a browser. These behaviors include browser interactions such as navigating to a web page and element interactions such as clicking a DOM element. Assertions are test code statements used to validate whether a DOM element's value or state meets certain conditions. For example, an assertion can be checking the presence of a DOM element on the current web page.

```
1 test('age', async () => {
2 const driver = new Builder().forBrowser('chrome').
      build();
3 driver.get('http://localhost:5000');
4 name = driver.findElement(By.id('name'));
5 name.sendKeys('Bob', Key.ENTER);
6 expect(driver.findElement(By.id('age').value).tobe
      (23);
7 driver.close();
8 }
```

**Listing 1: A web e2e UI test example (age.test.js).**

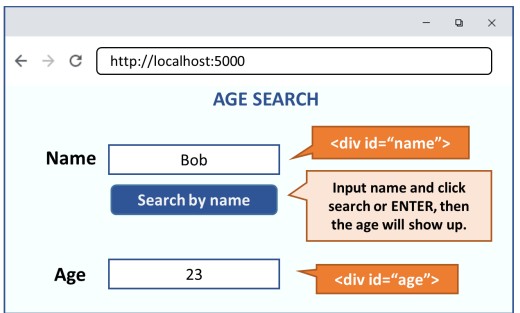

**Figure 1: The web page of the example shown in Listing 1.**

Listing 1 shows a web e2e UI test example written in JavaScript. It tests the web page shown in Figure 1 using the Selenium web driver [44]. Selenium is one of the most popular web drivers that provides various commands to interact with web elements similarly to a typical web user. Test code shown in Listing 1 first creates a `driver` instance (line 2) and navigates to the web page located at 'http://localhost:5000' (line 3). Next, the test code finds the element with id 'name' (line 4), adds a string 'Bob' into this editable field, and clicks the ENTER key (line 5). As shown in Figure 1, these commands will initiate a search for the age of the person named 'Bob'. Finally, the code asserts if the value of the element with id 'age' returned from the search is equal to the expected outcome 23 (line 6). If so, the test passes. Otherwise, the test fails.

### 2.2 UI-Based Flakiness

One common type of flakiness in web e2e UI test is caused by nondeterministic execution orders between the test code and the

client-side code under test. In web e2e testing, the test code sends commands through WebDriver to guide the client-side code execution, where each command may trigger multiple client-side operations. For example, clicking the ENTER key may result in several operations, including sending a value to the server, retrieving data from a database based on the input, and finally displaying the result on the web page. The time to complete all associated operations is subject to various factors, *e.g.,* network delays and server loads. If some operations from one command are not completed before executing the next command, the UI test can yield flaky results because of potential data or control flow dependencies between commands.

The test code shown in Listing 1 contains one such flakiness. The command `sendKeys` at line 5 inputs 'Bob' and initiates a server search for Bob's age. Once the search is done, the age value will be sent back to the client side and then inserted and displayed in an element with id 'age'. The time required to complete these changes can affect whether the element is updated before the subsequent assertion at line 6. If the update is successful, the test will pass; otherwise, it will fail. Therefore, this test appears to be flaky.

### 2.3 Flaky-Prone Commands

A command is deemed *flaky-prone* if its associated operations may not fully complete before proceeding to the subsequent command. We quantify the likelihood of a test being flaky by calculating the proportion of flaky-prone commands among all commands. Focusing on UI-based flakiness, we keep track of flaky-prone commands by monitoring all DOM changes a command causes. DOM changes are described as *mutations* in the browser.

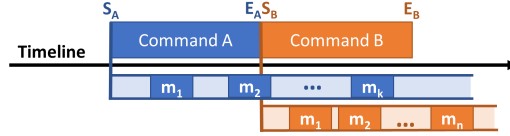

**Figure 2: Interleaved mutations triggered by two consecutive commands.**

Figure 2 shows two consecutive commands $A$, $B$, and their triggered mutations. In JavaScript, commands are implemented with promise: once the promise of command $A$ is settled, command $B$ initiates. Let $S_A$ and $E_A$ denote the start and promise settled time of command $A$, and $T_m$ represents the occurrence time of mutation $m$. For a command $C$ and its triggered mutation set $M$, if $\exists m \in M$ such that $T_m > E_C$ (*i.e.,* there exists a mutation that happens after the promise is settled), we say command $C$ is *flaky-prone*. In Figure 2, both command $A$ and $B$ are flaky-prone, since $T_{m_k} > E_A$ and $T_{m_n} > E_B$. Notably, $T_{m_k} > S_B$ indicates that mutations from command $A$ haven't concluded when $B$ starts to execute.

## 3 WEFIX

In this section, we describe WEFix, a tool that can automatically and intelligently insert explicit waits to fix UI-based flakiness. The workflow of WEFix is shown in Figure 3. WEFix consists of two components, *Mutation Recorder* and *Oracle Generator*. The goal of Mutation Recorder is to record runtime mutation events for each

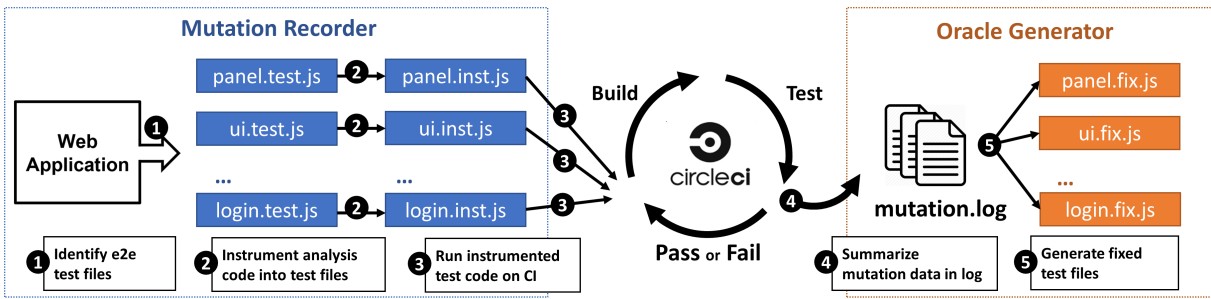

Figure 3: The workflow of WEFix.

command. Based on the mutations collected by Mutation Recorder, Oracle Generator generates a proper wait oracle for each command in the test code. Specifically, Mutation Recorder first identifies all e2e test files written in JavaScript in a given web application (❶). It then adds analysis code into the test files through instrumentation (❷). Next, it runs the instrumented code on a platform, *e.g.,* circleCI (❸). The next step is to record the runtime DOM mutations and save them in a log file (❹). Finally, the Oracle Generator uses the mutation records to generate proper fixes and transform the test files based on the fix (❺).

## 3.1 Mutation Recorder

Browsers provide a web interface called MutationObserver [35] that monitors changes made to the DOM tree. However, it is difficult to determine which mutation is triggered by which command. In addition, test frameworks do not provide reliable data transfer methods between the test code and the browser-side code under test. To address these challenges, Mutation Recorder is designed to collect mutation information at runtime for each command.

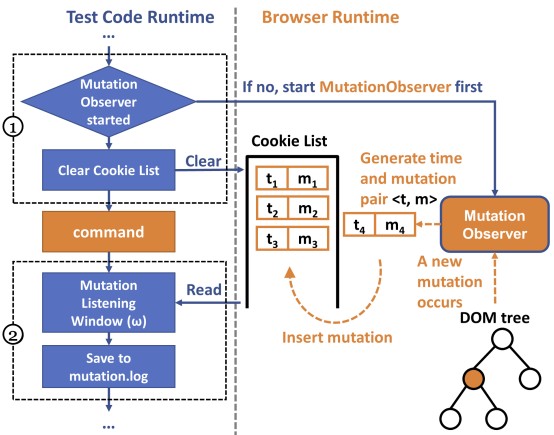

Figure 4: Mutation Recorder runtime.

Figure 4 shows the runtime of the Mutation Recorder. The left side of Figure 4 shows the runtime of a single command in the test code, and the right side shows the runtime of mutations in the browser.

### 3.1.1 Test Code Runtime.
On the test code side, Mutation Recorder instruments and adds analysis code before (①) and after (②) each command. To instrument each command, we parse the test code to an Abstract Syntax Tree (AST) using the Babel [8] library and identify command functions in the test code that perform browser interactions or element interactions. Before each command is executed (①), the analysis code first ensures the MutationObserver API is enabled. This step is needed because the MutationObserver API is turned off by default when navigating a new page. Next, the analysis code clears the cookie list used to store history mutations. To enable the test code to reliably generate wait oracles, we need to send mutations information observed from the browser side to the text code side on the fly. However, the current web drivers that use W3C wire protocol [54] for communication between the test code and the browser do not provide a reliable way to transmit extra data. Thus, Mutation Recorder innovatively resorts to a web mechanism, cookies, which are useful in temporarily storing data and can be read and modified by the web driver. During implementation, we add identifiable labels to our created cookies, ensuring that existing cookies are not cleared and other functionalities are preserved.

After the command is executed (②), the test code keeps listening for mutations by reading the cookie list for $\omega$ seconds. $\omega$ represents the listening window size. It is tricky to decide the value of $\omega$ in advance, *i.e.,* how much time needs to spend on listening. If $\omega$ is too small, not all mutations are fully recorded; if $\omega$ is too large, the runtime overhead would be greatly increased. Therefore, we design Algorithm 1 to dynamically adjust the window size $\omega$ in real time.

---

**Algorithm 1** Dynamic Listen Window

---

1: $T \leftarrow Time.now()$
2: $\omega \leftarrow 1$ {Initialize window size as 1 second}
3: **while** $Time.now() < T + \omega$ **do**
4:     **if** new mutation $m$ occurs **then**
5:         $RT \leftarrow m.time - T$ {m's relative time}
6:         $\omega \leftarrow \max(2 * RT, \omega)$
7:         $\omega \leftarrow \min(20, \omega)$ {$\omega$ not exceed 20}
8:     **end if**
9: **end while**

---

The main idea of this algorithm is to double the size of the listening window whenever a new mutation occurs to leave enough time for the next possible mutation. $T$ represents the start time of the listening (line 1) and window size $\omega$ is initialized as 1 (line 2). Then in the while loop, it keeps listening for mutations until $\omega$ seconds have passed. When a new mutation occurs, it first calculates

the mutation's *Relative Time RT*, which is defined as its occurrence time minus the start time $T$. Then $\omega$ will be set to $2 \times RT$. Meantime, $\omega$ should not decrease or exceed a maximum value. The maximum value, set empirically at 20, is utilized to ensure a sufficiently long waiting period to ensure that all mutations are recorded in our data. After the mutation listening stage, all the mutation records would be saved to a local file named *mutation.log*.

*3.1.2 Browser Runtime.* On the browser side, the runtime workflow is signified with the orange dotted line. Every time a new mutation occurs on the DOM tree, the MutationObserver will generate a mutation record $m$, constituting a *<t, m>* pair with the mutation occurrence time $t$. Then the newly built *<t, m>* pair will be inserted into the cookie list.

## 3.2 Oracle Generator

Explicit waits are preferred compared to implicit waits for fixing flaky e2e tests in the community [11, 25, 42]. The key of explicit waits is to determine expected conditions, which we call *oracles*. It requires in-depth knowledge of DOM element changes for developers to select accurate expected conditions as the oracle. Besides, multiple rounds of testing are usually needed to ensure that the oracle works both correctly and efficiently.

Listing 2 is an example of using explicit waits to fix the flakiness in Listing 1. A `wait-until` style statement (line 3) is added after the flaky-prone command (line 2). In this code example, Selenium will wait for a maximum of 4 seconds until the oracle stands. Internally, the oracle is checked every 100 milliseconds until it returns `True`.

```
1 ...
2 name.sendKeys('Bob', Key.ENTER);
3 WebDriverWait(driver, 4).until(EC.
      presence_of_element_located((By.id, 'age')))
4 expect(driver.findElement(By.id('age').value).tobe
      (23);
5 ...
```

**Listing 2: Explicit waits manually added to fix the flakiness in Listing 1.**

However, it is challenging to choose a proper wait oracle. If the waiting time is too long, it can impose substantial runtime overhead in CI. If the waiting time is too short, it may not be able to fix the flakiness at all. To ensure a proper wait oracle, Oracle Generator is designed to automatically generate an appropriate oracle for each command based on mutations collected by Mutation Recorder. Specifically, Oracle Generator first prunes irrelevant mutations (Sec. 3.2.1), then constructs mutation state machines (Sec. 3.2.2), generates oracles for each command (Sec. 3.2.3), and finally adds explicit waits in the text code (Sec. 3.2.4).

*3.2.1 Pruning Irrelevant Mutations.* To minimize the runtime overhead, Oracle Generator first prunes two types of mutations that are irrelevant to UI tests.

**GUI-Irrelevant Mutation.** Not all mutations result in GUI changes. Since our research focuses on UI-based flaky tests, we prune GUI-irrelevant mutations from the collected mutations. Specifically, we detect and remove three types of GUI-irrelevant mutations: (1) Mutations outside the HTML document <body> (*e.g.,* <head> and <meta> elements) since these changes are not observable to a web user; (2) Mutations that do not change the Cascading Style Sheet (CSS) style of any element; (3) Mutations on target elements that are invisible to the user.

**Background Mutation.** Some web pages may generate mutations in the background periodically, even without any user interaction. Background mutation is common in web page image rotation and animation. As background mutations are not triggered by any command in the test code, it's important to filter them out. To do so, we aggregate all mutations on a web page and detect background mutations based on their triggering source and occurrence time.

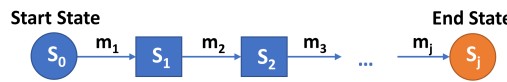

**Figure 5: Finite state machine (FSM) representing DOM tree status transitions via mutations $m_1, m_2, ..., m_j$.**

*3.2.2 Mutation State Machine.* After pruning irrelevant mutations, we construct a finite-state machine (FSM) to model the DOM mutations triggered by each command. Assume the mutation list for one command is represented as $(m_1, m_2, ..., m_j)$, where mutations are ordered by their occurrence time. As shown in Figure 5, the states in this FSM are DOM status at different times, and the transition between states is mutation $m_i$. The start state $S_0$ represents the DOM status right after the command is executed. The end state $S_j$ represents the DOM status when no more mutation occurs.

Assume there are $n$ elements on the DOM tree. We define the FSM state $S_i$ ($0 \leq i \leq j$) as the set of all DOM tree element's status:

$$S_i := \{E_1, E_2, ..., E_n\} \tag{1}$$

where $E_i$ ($0 \leq i \leq n$) represents the $i$-th element's status:

$$E_i := \{A_1^0, A_2^0, ..., A_k^0, T^0, C^0\} \tag{2}$$

The element status is the set of all properties that are subject to GUI changes during runtime. Assume the element has $k$ attributes, $A_1, A_2, ..., A_k$. $T$ is the element's text value. $C$ is the element's child list size. Each component is marked with a superscript, which represents the index of the mutation deriving the element's status. In the start state, the superscript is set to 0, meaning that there are no mutations yet. When a mutation $m_t$ occurs and changes the value of a property $P^0$, the property will be updated to $P^t$. Adding a superscript differentiates element status with identical properties, ensuring the state machine proceeds linearly.

*3.2.3 Oracle Generation Algorithm.* The goal of Oracle Generator is to generate a wait oracle determining the end state of the mutation finite state machine. When the oracle is met, no further mutation will occur. Without loss of generality, Oracle Generator uses three properties with the latest mutation times in the end state $S_j$ (the properties may come from different elements) to generate three oracles and combine them together to represent the end state. The oracle generation algorithm is shown in Algorithm 2.

**Algorithm 2** Generate Oracle

**Input:** end state: $S_j$
**Output:** oracle: $O$
1: Initialization: $O \leftarrow True$
2: $P_1, P_2, P_3 \leftarrow$ properties with largest timestamp in $S_j$
3: **for** $P$ in $[P_1, P_2, P_3]$ **do**
4:     $Xpath \leftarrow P.element.toXpath()$
5:     **if** $P$ is $A$ **then**
6:         $oracle \leftarrow$ "$get(\${Xpath}).should(have.\${P.Name}, \${P.Value})$"
7:     **end if**
8:     **if** $P$ is $T$ **then**
9:         $oracle \leftarrow$ "$get(\${Xpath}).should(have.text, \${P.Value})$"
10:     **end if**
11:     **if** $P$ is $C$ **then**
12:         $oracle \leftarrow$ "$get(\${Xpath}.).child.should(have.len, \${P.Value})$"
13:     **end if**
14:     $O \leftarrow O \ \& \ oracle$
15: **end for**

The algorithm takes the end state $S_j$ as input and outputs an oracle $O$. The algorithm selects three properties $P_1$, $P_2$, and $P_3$, which have the latest mutation time in the end state $S_j$. The number of selected properties is a trade-off between stability and complexity: too few may not remove flakiness, while too many could generate excessively long statements, compromising human readability and inducing additional runtime overhead. In our dataset, selecting three properties balances a high fix rates with readability and low overhead. For each selected property, the algorithm creates a Xpath [36] that the web driver can utilize to fetch the element from the DOM tree [43]. Next, it creates an oracle using the value of the property, where the property's type can be attributes ($A$), text ($T$), or child list size ($C$). Finally, three oracles are combined with a logical AND (&), meaning that all three oracles should be met.

*3.2.4 Adding Explicit Waits.* The last step is to insert the generated oracle as an explicit wait in the original test code. Listing 3 shows the inserted wait to the command in Listing 1.

```
1 ...
2 name.sendKeys('Bob', Key.ENTER);
3 waitUntil(driver.get("//*[@id='age']").should(have.
      text, 23));
4 ...
```

**Listing 3: Explicit waits automatically created by WEFix to fix the flakiness in Listing 1 (age.fix.js).**

In the fixed code, Oracle Generator adds an assertion after entering a name Bob and clicking the ENTER key. As the `waitUntil` function will suspend the test process until the oracle is met, the test waits until the age element returns a value of 23 after pressing the ENTER key.

## 4 EVALUATION

We first construct a dataset of UI-based flaky tests from real-world GitHub projects. We extensively evaluate WEFix on these real-world projects for its effectiveness and efficiency. We also compare WEFix with implicit wait approaches commonly used in practice. In particular, our evaluation aims to answer four important research questions:

- **RQ1 (UI-Based Flakiness Prevalence):** How many commands are flaky-prone in real-world projects?
- **RQ2 (Implicit Wait Overhead):** What is the performance overhead of implicit waits for fixing UI-based flakiness?
- **RQ3 (WEFix Effectiveness):** How many UI-based flakiness can be effectively fixed by WEFix?
- **RQ4 (WEFix Efficiency):** What is the runtime overhead of WEFix and how does it compare with other approaches?

### 4.1 Dataset

Constructing a dataset of real-world UI-based flaky tests is nontrivial [30]. We collected 100,000 popular JavaScript repositories from GitHub using GitHub Search API [21]. In order to find out projects containing web e2e tests, we iterate through each project's dependency file *package.json*, and use the following JavaScript testing framework names as keywords to search for projects containing web e2e tests: *selenium*, *cypress*, *testcafe*, *nightwatch*, *protractor*, *playwright*, *webdriverio* and *webdriver*. This search resulted in 250 repositories, and the list is available on the project web page [6]. It is noteworthy that repositories containing e2e tests are relatively rare, primarily because e2e testing is more frequently used in commercial web products than in open-source projects. We believe that our curated dataset of open-source projects with e2e tests can present a valuable resource for future research in web e2e tests.

To ensure quality and popularity, we focus on the 37 repositories that have over 20,000 stars. Some repositories' tests are not deployable due to the absence of test documentation or local execution conditions, some of which require online CI environments with server-side support. From those that are operational, We randomly select six projects and intentionally include the *keystone* project (8,300 stars) to represent repositories with fewer than 20,000 stars. These seven projects are used widely by organizations in the real world. For example, Storybook [22] is a prevalent tool for UI development, adopted by projects owned by notable companies like Microsoft, Shopify, Airbnb, and Salesforce [51]. Keystone [20], a Content Management System (CMS), is employed by organizations including Atlassian and Csiro for their software development.

### 4.2 Implementation

WEFix is implemented in JavaScript as an NPM package [7]. Developers can integrate WEFix into their web application for e2e testing in Node.js. In addition, our tool provides a user-friendly GUI to visually present collected mutation records, facilitating the analysis of the flakiness. The source code and dataset can be found on GitHub [1] and archived on Figshare [4, 6]. Deployment details are available on the GitHub page [5].

**Table 1: Mutation relative time (RT) result.**

| Repo | Star | avg. RT(ms) | avg. latest RT(ms) | %flaky-prone |
|---|---|---|---|---|
| storybookjs/storybook | 80.6k | -29 | 564 | 39.9% |
| nolimits4web/swiper | 36.8k | -13 | 790 | 67.1% |
| carbon-app/carbon | 33.3k | 62 | 1034 | 78.5% |
| atlassian/react-beautiful-dnd | 31k | 69 | 251 | 89.7% |
| react-hook-form/react-hook-form | 37k | -12 | 470 | 80.8% |
| getredash/redash | 23.9k | -11 | 619 | 62.7% |
| keystonejs/keystone | 8.3k | 17 | 201 | 41.2% |
| AVERAGE | | 7 | 561 | 65.7% |

## 4.3  RQ1: UI-Based Flakiness Prevalence

We investigate the prevalence of UI-based flakiness in real-world web e2e tests. To do so, we run WEFix on all 367 e2e tests from the seven selected projects. To better represent flaky-prone commands, we report the Relative Time (RT) of a mutation, denoted as its occurrence time relative to the next command. A negative RT means the mutation occurs before the next command, and vice versa. The result is shown in Table 1. The column "avg.RT" is the average relative time (RT) of all mutations collected in one project. "avg. latest RT" is the average RT of the last mutation triggered by each command in the project. "%flaky-prone" refers to the proportion of flaky-prone commands among all commands.

In Table 1, the "avg.RT" is distributed within a range of 0 ± 100 ms. As can be seen, three projects have a positive "avg.RT", indicating that UI-based flakiness is common. The "avg.latest RT" is 561. If we choose to wait until the last mutation occurs, the average wait time needed is around 561 ms. Besides, we count the percentage of commands with positive RTs. We find that the average "%flaky-prone" is 65.7%, and the *atlassian/react-beautiful-dnd* project has the highest "%flaky-prone" of 89.7%.

> **RQ1 Takeaway:** On average, 65.7% of the commands in the studied real-world UI tests are flaky-prone.

## 4.4  RQ2: Implicit Wait Overhead

We investigate performance overhead of adding implicit waits after each command. Listing 4 demonstrates adding a 2-seconds wait following the `sendKeys` command in Listing 1, allowing the web page to display the age retrieved from the server.

```
1 ...
2 name.sendKeys('Bob', Key.ENTER);
3 driver.waitFor(2) # seconds
4 expect(driver.findElement(By.id('age').value).tobe
    (23);
5 ...
```

**Listing 4: A 2-second implicit wait added after all commands.**

Figure 6 illustrates the cumulative distribution of RT, measured in milliseconds, across seven projects. It presents the proportion of mutations whose RT is less than or equal to the value on the X-axis. Based on the plot, over 95% of mutations' RT is less than 2,000 ms (2 second). This observation informs a straightforward strategy to mitigate UI-based flakiness: adding a 2-seconds wait after each command in the test code so that most mutations can finish within this period.

Although adding a 2-seconds wait is simple to implement, it introduces significant runtime overhead. Table 2 shows the performance comparison between the original test suite and instrumented implicit wait version. The 2-second wait approach takes 2.4× to 4.5× time compared to the original test code. For instance, the project *redash* increases the test time from 17 minutes to 41 minutes across its 41 test files after adding implicit waits. This dramatic surge in test time, especially in a Continuous Integration (CI) environment where tests run upon each new commit, can significantly slow down software development iteration.

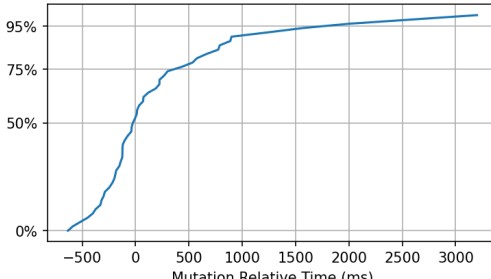

**Figure 6: Cumulative distribution of mutation RT across seven projects.**

**Table 2: Test time comparison of the original test and 2-second implicit wait.**

| Repo | # test file | Test Time | |
|---|---|---|---|
| | | original | 2-seconds wait |
| storybookjs/storybook | 2 | 16s | 44s (3.1x) |
| nolimits4web/swiper | 5 | 20s | 1m30s (4.5x) |
| carbon-app/carbon | 7 | 40s | 2m58s (4.5x) |
| atlassian/react-beautiful-dnd | 7 | 16s | 1m10s (4.4x) |
| react-hook-form/react-hook-form | 35 | 3m03s | 8m17s (2.7x) |
| getredash/redash | 41 | 17m04s | 41m10s (2.4x) |
| keystonejs/keystone | 5 | 29s | 2m05s (4.3x) |

> **RQ2 Takeaway:** Adding a 2-second wait after commands introduces significant runtime overhead (2.4× to 4.5×).

## 4.5  RQ3: WEFix Effectiveness

To evaluate WEFix effectiveness, we first reproduce the UI-based flaky tests in the seven projects. Unfortunately, it is widely believed in the field that web e2e flaky tests are difficult to reproduce. Web e2e testing requires multiple parts of an application to work together correctly, with complicated setup processes that differ from application to application and even from version to version. Besides, the ability to reproduce flaky tests is also affected by device, network, and other environmental factors.

To address these challenges, we reproduce UI-based flakiness inspired by developers' actual fix. In practice, developers typically fix UI-based flakiness by manually inspecting its root cause and inserting wait statements in the appropriate locations within the test code. Based on this observation, we reproduce the flakiness by removing these wait statements added by developers and then rerunning these tests. Given that WEFix is designed to help developers automatically inject explicit waits to mitigate flakiness, our method to reproduce flakiness closely mirrors real-world development scenarios.

Rerunning a test multiple times to see if all of them pass is one of the most effective methods to test whether a test is flaky. Existing testing frameworks [14] normally support reruns, *e.g.,* 3, 5, and 10 reruns. We follow the common practice used by existing work [47] that rerun each test ten times. If all 10 runs pass, there is a high probability that the test is not flaky. If any failure appears during the rerun, we mark the test as *c-flaky*. Among the 367 tests, we

**Table 3: Efficiency and effectiveness of WEFix vs. implicit waits approaches.**

| repo | test file | original run time(s) | WEFix overhead | # fixed | 0.2-seconds Fix overhead | # fixed | 0.5-seconds Fix overhead | # fixed | 1-second Fix overhead | # fixed | 2-seconds Fix overhead | # fixed |
|---|---|---|---|---|---|---|---|---|---|---|---|---|
| storybook | navigation.spec.ts | 11.21 | 16.9% | 1 | 35.2% | 0 | 54.0% | 1 | 85.2% | 1 | 147.6% | 1 |
| swiper | a11y.js | 1.23 | 2.8% | 2 | 82.1% | 1 | 179.7% | 1 | 342.3% | 2 | 667.5% | 2 |
| | core.js | 4.85 | 36.3% | 1 | 1593.2% | 0 | 1754.0% | 1 | 2022.1% | 1 | 2558.1% | 1 |
| carbon | background-color.spec.js | 6.52 | 21.0% | 1 | 111.0% | 1 | 198.5% | 1 | 344.2% | 1 | 635.6% | 1 |
| | basic.spec.js | 4.26 | 7.5% | 1 | 66.7% | 1 | 137.1% | 1 | 254.5% | 1 | 489.2% | 1 |
| | security.spec.js | 12.24 | 10.8% | 3 | 21.8% | 0 | 31.6% | 0 | 48.0% | 2 | 80.6% | 3 |
| react -beautiful -dnd | focus.spec.js | 1.25 | 50.4% | 4 | 450.4% | 4 | 978.4% | 4 | 1858.4% | 4 | 3618.4% | 4 |
| | move-between-lists.spec.js | 0.85 | 20.0% | 1 | 112.9% | 1 | 254.1% | 1 | 489.4% | 1 | 960.0% | 1 |
| | reorder-lists.spec.js | 0.89 | 23.6% | 1 | 104.5% | 1 | 239.3% | 1 | 464.0% | 1 | 913.5% | 1 |
| | reorder-virtual.spec.js | 0.86 | 17.4% | 0 | 125.6% | 0 | 265.1% | 0 | 497.7% | 0 | 962.8% | 1 |
| | reorder.spec.js | 0.87 | 6.9% | 1 | 94.3% | 0 | 232.2% | 1 | 462.1% | 1 | 921.8% | 1 |
| react-hook -form | autoUnregister.ts | 2.45 | 9.4% | 1 | 79.2% | 1 | 164.9% | 1 | 307.8% | 1 | 593.5% | 1 |
| | basic.ts | 17.17 | 16.5% | 4 | 105.0% | 4 | 222.1% | 4 | 417.2% | 4 | 807.4% | 4 |
| | basicSchemaValidation.ts | 13.63 | 10.8% | 3 | 118.0% | 3 | 254.5% | 3 | 482.0% | 3 | 936.8% | 3 |
| | conditionalField.ts | 2.69 | -2.7% | 1 | 119.7% | 1 | 220.1% | 1 | 387.4% | 1 | 721.9% | 1 |
| | controller.ts | 5.2 | 17.3% | 3 | 133.3% | 0 | 277.5% | 3 | 517.9% | 3 | 998.7% | 3 |
| | customSchemaValidation | 13.45 | 27.9% | 3 | 110.5% | 3 | 248.8% | 3 | 479.3% | 3 | 940.2% | 3 |
| | reValidateMode.ts | 12.9 | 16.4% | 6 | 10.6% | 5 | 71.1% | 5 | 171.9% | 6 | 373.4% | 6 |
| | formState.ts | 9.43 | 31.2% | 6 | 204.9% | 5 | 398.9% | 5 | 722.4% | 5 | 1369.2% | 6 |
| | formStateWithNestedFields.ts | 8.56 | 20.8% | 6 | 148.8% | 5 | 306.5% | 6 | 569.4% | 6 | 1095.1% | 6 |
| | formStateWithSchema.ts | 9.59 | 27.3% | 6 | 148.9% | 5 | 305.3% | 6 | 566.0% | 6 | 1087.4% | 6 |
| | isValid.ts | 4.08 | 7.6% | 4 | 148.0% | 4 | 295.1% | 4 | 540.2% | 4 | 1030.4% | 4 |
| | manualRegisterForm.ts | 4.97 | 4.8% | 1 | 94.8% | 1 | 203.4% | 1 | 384.5% | 1 | 746.7% | 1 |
| | reset.ts | 2.19 | 4.6% | 1 | 75.8% | 1 | 171.7% | 1 | 331.5% | 1 | 651.1% | 1 |
| | useFieldArray.ts | 21.37 | 24.9% | 9 | 159.3% | 8 | 385.3% | 9 | 762.0% | 9 | 1515.4% | 9 |
| | setError.ts | 2.18 | -1.8% | 2 | 106.0% | 2 | 216.1% | 2 | 399.5% | 2 | 766.5% | 2 |
| | useFormState.ts | 5.27 | 1.1% | 1 | 124.1% | 1 | 255.0% | 1 | 473.2% | 1 | 909.7% | 1 |
| | useWatch.ts | 2.76 | 12.3% | 3 | 87.3% | 3 | 217.8% | 3 | 435.1% | 3 | 869.9% | 3 |
| | watch.ts | 2.22 | 0.5% | 1 | 123.9% | 1 | 218.5% | 1 | 376.1% | 1 | 691.4% | 1 |
| redash | dashboard_spec.js | 7.83 | 13.8% | 1 | 57.5% | 1 | 134.1% | 1 | 261.8% | 1 | 517.2% | 1 |
| | view_alert_spec.js | 6.28 | 27.1% | 3 | 64.5% | 3 | 117.0% | 3 | 204.6% | 3 | 379.8% | 3 |
| | filters_spec.js | 3.6 | 1.4% | 0 | 47.5% | 0 | 97.5% | 0 | 180.8% | 1 | 347.5% | 1 |
| | query/parameter_spec.js | 30.78 | 14.1% | 13 | 71.9% | 9 | 157.6% | 11 | 300.6% | 12 | 586.5% | 13 |
| | sharing_spec.js | 11.36 | 7.9% | 2 | 110.4% | 0 | 134.2% | 0 | 173.8% | 2 | 253.0% | 2 |
| | create_data_source_spec.js | 5.4 | 3.5% | 3 | 93.5% | 3 | 171.3% | 3 | 300.9% | 3 | 560.2% | 3 |
| | create_destination_spec.js | 3.05 | 24.3% | 2 | 81.0% | 2 | 140.0% | 2 | 238.4% | 2 | 435.1% | 2 |
| | share_embed_spec.js | 10.55 | 47.6% | 1 | 49.5% | 1 | 117.7% | 1 | 231.5% | 1 | 459.0% | 1 |
| | filters_spec.js | 3.59 | 36.2% | 2 | 128.7% | 2 | 279.1% | 2 | 529.8% | 2 | 1031.2% | 2 |
| | dashboard/parameter_spec.js | 7.58 | 20.3% | 4 | 54.2% | 0 | 109.6% | 0 | 202.0% | 4 | 386.7% | 4 |
| | organization_settings_spec.js | 3.83 | 0.0% | 1 | 77.3% | 1 | 139.9% | 1 | 244.4% | 1 | 453.3% | 1 |
| | edit_profile_spec.js | 7.67 | 43.3% | 5 | 48.6% | 0 | 111.2% | 5 | 215.5% | 5 | 424.1% | 5 |
| | box_plot_spec.js | 2.19 | 0.0% | 1 | 42.0% | 1 | 83.1% | 1 | 151.6% | 1 | 288.6% | 1 |
| | choropleth_spec.js | 5.09 | 3.9% | 1 | 58.0% | 1 | 116.9% | 1 | 215.1% | 1 | 411.6% | 1 |
| keystone | filters.test.ts | 4.67 | 19.5% | 1 | 62.5% | 1 | 113.9% | 1 | 199.6% | 1 | 370.9% | 1 |
| | list-view-crud.test.ts | 8.97 | 12.7% | 3 | 55.0% | 2 | 95.1% | 3 | 162.0% | 3 | 295.8% | 3 |
| **SUMMARY** | | | **16.0%** | **120** | **133.3%** | **89** | **241.7%** | **106** | **422.3%** | **118** | **783.6%** | **122** |

reproduce 122 tests exhibiting flakiness and use these reproduced c-flaky tests for the effectiveness evaluation.

Table 3 presents the WEFix fix result on the 122 c-flaky tests. The second column presents the e2e test files containing c-flaky tests. To verify the effectiveness of the WEFix, we compare it with four implicit wait methods, each with different waiting times: 0.2s, 0.5s, 1s, and 2s. We measure the percentage of test files that could be successfully fixed. If the repaired test passes 10 reruns, we mark it as *fixed*, the number of which is shown in "# fixed" columns.

WEFix successfully fixes 120 out of the 122 tests, achieving a 98.4% fix rate. In comparison, implicit wait methods fix 89, 106, 118, and 122 tests, respectively, revealing an upward trend in the fix rate with increasing wait time. In Table 3, failed fix cases are shown with blue boxes. WEFix failed to fix two test files. Our analysis shows that the flakiness in both cases is caused by inherent flakiness

introduced by third-party tools that WEFix cannot access and thus cannot generate a proper fix for them.

> **RQ3 Takeaway:** WEFix successfully fixes 98.4% the UI-based flakiness, outperforming implicit wait methods of 0.2 seconds, 0.5 seconds, and 1 second.

## 4.6 RQ4: WEFix Efficiency

One major design goal of WEFix is to incur a low runtime overhead while ensuring a high fix rate. We collect the run times of the original test, test code applying WEFix, and code with implicit waits after all commands. We compute the overhead for each method as the percentage increase in run time introduced by the method. The result is shown in Table 3. Compared with our tool, implicit wait methods have much larger overheads. The 0.2-sec, 0.5-sec, 1-sec,

and 2-sec methods have an average overhead of 2.33× (133%), 3.42× (242%), 5.22× (422%), and 8.84× (784%), respectively.

**Table 4: One round e2e test time of seven projects.**

| Repo | Original | WEFix | Implicit Wait | | | |
|---|---|---|---|---|---|---|
| | | | 0.2 sec | 0.5 sec | 1 sec | 2 sec |
| storybook | 14s | 18s | 17s | 22s | 29s | 44s |
| swiper | 20s | 25s | 27s | 38s | 55s | 1m30s |
| carbon | 40s | 47s | 54s | 1m15s | 1m49s | 2m58s |
| react-beautiful-dnd | 16s | 18s | 21s | 30s | 43s | 1m10s |
| react-hook-form | 3m03s | 3m52s | 3m34s | 4m22s | 5m40s | 8m17s |
| redash | 17m04s | 20m47s | 19m29s | 23m06s | 29m07s | 41m10s |
| keystone | 29s | 41s | 39s | 53s | 1m17s | 2m05s |
| **Overhead** | | **1.25×** | **1.27×** | **1.68×** | **2.35×** | **3.7×** |

We also measure project-level overhead. We apply these methods to the entire e2e test suite of each project and compare the duration of one round of e2e testing. The result is shown in Table 4. WEFix has the lowest average overhead (1.25×), while the 2-sec wait method incurs the highest (3.7×).

> **RQ4 Takeaway:** Compared with implicit wait approaches, WEFix significantly reduces the runtime overhead while ensuring a high fix rate.

## 5 DISCUSSION

Although our experiments have shown that WEFix is both efficient and effective, WEFix has several limitations. First, our tool is configured to support e2e tests using either Cypress or Selenium, two of the most widely used e2e testing frameworks [12]. However, our current implementation does not support other e2e testing frameworks. We believe that WEFix can be extended to any e2e testing framework with adding additional support for their grammars.

Second, while the GitHub projects in our dataset are broadly used by real-world organizations, they are relatively small web development tools, compared with large, mature web applications or frameworks, to which we currently have no access. New challenges may arise when WEFix is applied to those large web applications with a lot more e2e tests. Moving forward, we plan to reach out to those organizations managing such large-scale applications, aiming to collaborate and explore the applicability of WEFix in more extensive web environments.

## 6 RELATED WORK

**Flaky Test Studies.** Flaky tests have been extensively studied. Luo et al. [33] investigated 201 commits from 51 projects that fixed flaky tests and classified flakiness into several types, including concurrency problems. Eck et al. [16] presented an empirical study of flaky tests from the perspective of developers, firstly introducing "test case timeout" category flakiness, which requires surprising effort to fix. Lam et al. [29] categorized flakiness from pull requests in six Microsoft subject projects. Gruber et al. [24] performed an empirical analysis of flaky tests in Python. Parry et al. [39] provided a comprehensive survey of 76 papers on flaky test research.

**Flaky Test Detection.** Armed with basic knowledge of flakiness, many works have been done to detect flaky tests. Based on historical commits in open-source projects, Luo et al. [33] offered insights into manifesting test flakiness. Bell et al. [10] presented an approach using differential coverage for detecting flaky tests. Shi et al. [46] presented a tool targeting flakiness arising from deterministic implementations of non-deterministic specifications. Silva et al. [49] focus on detecting *asynchronous wait* and *concurrency* flakiness by introducing environment noise using stress-loading tool. And many works [3, 15, 27, 40, 53] have been done to apply machine learning for flakiness detection.

**Mitigation and Repair.** There is increasing research efforts to mitigate or repair specific types of flaky tests. Bell et al. [9] presented a tool to mitigate flakiness due to order-dependent tests. Lam et al. [31] proposed an algorithm to enhance the soundness of test suite prioritization with respect to test order dependency. Shi et al. [45] investigates how to mitigate inconsistent coverage during mutation testing and developed iFixFlakies [47], a tool for automatically repairing order-dependent tests. Fazzini et al. [18] developed a tool to automatically generate test mocks [50] for mobile applications. Zhang et al. [55] proposed a technique for repairing flaky tests related to non-deterministic specifications.

**Web E2e Flaky Tests.** In recent years, web e2e testing has received increasing attention from the academic community. Romano et al. [41] conducted an empirical study on UI flaky tests and found that "async wait" (i.e., concurrency) flakiness predominates in UI testing. Olianas et al. [38] provided an experience report of fixing web e2e flakiness from the industry perspective. Olianas et al. [37] proposed SleepReplacer, which replaces existing thread sleeps with explicit waits. However, SleepReplacer differs from WEFix in two aspects. First, it operates in a passive manner, only replacing existing thread sleeps, which inherently restricts its applicability. By contrast, WEFix can proactively insert explicit waits for flaky commands, making it applicable to any newly written e2e test. WEFix can also insert wait statements automatically during development time, thereby significantly reducing the manual effort to select proper explicit waits. Second, SleepReplacer requires multiple runs of the test to tentatively replace existing thread sleeps and validate no new flakiness is introduced, which takes several minutes to replace a single wait. WEFix generates the DOM mutation profile in a single run and only requires around one second on average to instrument one flaky command.

## 7 CONCLUSION

We present WEFix, an efficient tool that generates explicit waits for UI-based flakiness in web e2e testing. We evaluate WEFix on 122 UI-based flaky tests collected from seven popular GitHub web projects. Our evaluation shows that adding implicit waits after commands introduces significant runtime overhead. Compared with implicit wait approaches, WEFix significantly reduces the runtime overhead while still ensuring a high fix rate. In practice, WEFix can be applied to reduce runtime overhead for web e2e tests or assist developers in automatically and intelligently inserting wait statements during development, significantly reducing manual efforts.

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
