# OpenReview forum: "WEFix: Intelligent Automatic Generation of Explicit Waits for Efficient Web End-to-End Flaky Tests"
_ACM.org/TheWebConf/2024/Conference — TheWebConf24_

### Official Review · Reviewer_6eJn · 2023-10-30

**Novelty:** 4
**Technical Quality:** 4

**Review:**

# Summary
This paper proposes WEFix, a technique that can automatically generate explicit waits for web end-to-end (e2e) tests that are flaky due to UI changes. WEFix records the DOM mutations triggered by each command and generates proper wait oracles based on them. WEFix also uses a finite state machine to model the DOM mutation events and ensure the correctness and efficiency of the wait oracles. The paper evaluates WEFix on seven real-world web projects and shows that it can fix 98% of the UI-based flaky tests with low overhead.

# Pro
1.  This paper discusses an interesting yet important problem
2.  The paper evaluates WEFix on seven popular real-world web projects and shows that it can fix 98% of the UI-based flaky tests with low overhead
3.  The dataset is publicially available.

# Cons
1. Lack of comparison with other techniques
2. Limited evaluation on different browsers and platforms
3. Potential false positives and false negatives
4. Lack of user study or feedback


# Detailed Comments

1.  **Lack of comparison with other techniques**: The paper does not compare WEFix with other existing techniques for fixing UI-based flakiness, such as Flakiness Diagnosis and Fixing or Flakiness Detection and Localization. A comparison with these techniques could show the advantages and disadvantages of WEFix in different scenarios.
2. **Limited evaluation on different browsers and platforms**: The paper only evaluates WEFix on Chrome browser and CircleCI platform. However, UI-based flakiness may vary across different browsers and platforms, such as Firefox, Safari, or GitHub Actions. A more comprehensive evaluation on different browsers and platforms could demonstrate the robustness and generalizability of WEFix.
3. **Potential false positives and false negatives**: The paper does not discuss the possibility of false positives and false negatives in WEFix. False positives refer to cases where WEFix adds unnecessary explicit waits that do not fix any flakiness but increase the runtime overhead. False negatives refer to cases where WEFix fails to add explicit waits that are needed to fix the flakiness. A discussion on how to measure and reduce the false positives and false negatives could improve the reliability and accuracy of WEFix.
4. **Lack of user study or feedback**: The paper does not report any user study or feedback on WEFix. User study or feedback could provide valuable insights into how developers perceive and use WEFix, what are the benefits and challenges of using WEFix, and what are the potential improvements or extensions of WEFix.
5. **Lack of scalability evaluation**: The paper does not evaluate the scalability of WEFix on large-scale web applications or test suites. It is possible that WEFix may encounter performance issues or limitations when dealing with complex web applications or test suites that involve a large number of commands, mutations, elements, or properties. A scalability evaluation could show the practicality and applicability of WEFix in real-world scenarios.
6. **Lack of in-depth discussion on results**: This paper could benefit from a more in-depth discussion of the results.

**Questions:**

1. it’s unclear how this algorithm deals with complex scenarios where multiple mutations occur simultaneously or in a rapid sequence. Could you elaborate on how your

**Ethics Review Description:**

/

**Reviewer Confidence:**

3: The reviewer is confident but not certain that the evaluation is correct

**Scope:**

4: The work is relevant to the Web and to the track, and is of broad interest to the community

---

### Official Review · Reviewer_nrSK · 2023-11-12

**Novelty:** 5
**Technical Quality:** 5

**Review:**

This paper presents WEFix to automatically generate fix code for UI-based flakiness. It inserts explicit waits, which wait until certain condition is met, to mitigate flaky instances. The paper reads smoothily and I can follow its idea even though I am not an expert in web e2e test. The evaluation shows that the solution can significantly improve the efficiency of implicit wait solutions.

pros:
- it automatically inserts fix to mitigate flakiness
- the fix can adaptably adjust to web events

cons:
- values determined by experience (e.g., three properties and window size) may have significant influence on performance

**Questions:**

- about the originality: Is there any existing solutions that use explicit waits?
    In section 3.2 (page 4), it mentions some existing explicit waits. What about their performancce? why not compare them?

-  How to differentiate between flaky events and random events? some events may contain random values.

- Will the finite state machine be too large? If yes, what's the influence on the performance?

**Reviewer Confidence:**

3: The reviewer is confident but not certain that the evaluation is correct

**Scope:**

4: The work is relevant to the Web and to the track, and is of broad interest to the community

---

### Official Review · Reviewer_Pugt · 2023-11-20

**Novelty:** 5
**Technical Quality:** 6

**Review:**

*Overview*:

This paper proposes a framework for mutating flaky tests into reliability by automatically adding waits on explicit events (such as key values and so on).

*Clarity*:

The paper is well written and I could follow their arguments throughout. The literature review seems comprehensive and all figures and tables can be understood.

*Qualities/Strengths*:

This work seems extremely useful, and I am pleased to see that the authors have also released it as open source. The evaluation methodology over a number of open source projects seems sound, and the results are convincing.

*Weaknesses*:

No comparison with other approaches than just manually inserted delays.

**Questions:**

1. Why did the two failing cases fail? This is a small number and so the root cause could have been provided.

2. Are there cases where WEFix added unnecessary delays?

**Reviewer Confidence:**

3: The reviewer is confident but not certain that the evaluation is correct

**Scope:**

4: The work is relevant to the Web and to the track, and is of broad interest to the community

---

### Official Review · Reviewer_ojvd · 2023-11-21

**Novelty:** 5
**Technical Quality:** 6

**Review:**

**Quality:**

High Quality: The paper is well-structured, presenting a clear problem statement, detailed methodology, thorough evaluation, and insightful discussion. The research design is rigorous, and the results are supported by extensive data and analysis.

**Clarity:**

Clear and Comprehensive: The paper is well-written and easy to follow. Technical concepts are explained clearly, making it accessible to readers with a basic understanding of web testing. The use of figures, tables, and code snippets enhances understanding.

**Originality:**

Highly Original: The approach of using a Mutation Recorder and Oracle Generator to automatically insert explicit waits is novel. The idea of leveraging browser UI changes to predict client-side code execution is innovative and shows a deep understanding of the problem domain.

**Significance:**

Highly Significant: The paper addresses a critical and long-standing issue in web development - UI-based flakiness in e2e tests. The solution has the potential to significantly improve the reliability and efficiency of web testing, which is crucial for modern web application development.

**Pros:**

1. Innovative Approach: The use of explicit waits to handle flakiness is a significant improvement over traditional implicit wait approaches.

2. Effective Solution: WEFix demonstrates high effectiveness in fixing flaky tests, with a correctness of 98%.

3. Efficiency: The approach significantly reduces the runtime overhead compared to implicit waits.

4. Practicality: The tool is implemented as an NPM package, making it easily integrable into existing web development workflows.

5. Strong Evaluation: The paper presents a comprehensive evaluation using real-world projects, enhancing the credibility of the results.

**Cons:**

1. The innovative use of cookies for information storage in your paper is commendable for its ingenuity. However, the paper could benefit from more detailed explanations of certain technical terms, such as 'mutation' and 'oracle.' These terms, while well-understood within the domain, might be confusing to readers not specialized in web testing or those familiar with similar terms in the context of fuzzing. Clarifying these terms would enhance the paper's accessibility and prevent potential misunderstandings among a broader audience.

2. Dependence on Accurate Mutation Recording: The effectiveness of WEFix relies heavily on the accurate recording of DOM mutations, which might not always be reliable in every testing scenario.

3. Potential for Overhead in Certain Cases: While generally efficient, there might be scenarios where the dynamic adjustment of the wait window could lead to performance overhead.

**Questions:**

1. In your paper, the effectiveness of WEFix is significantly dependent on the accurate recording of DOM mutations. Could you elaborate on how WEFix ensures the reliability of this mutation recording across various web testing scenarios? Additionally, are there any particular scenarios where this mutation recording might be less reliable, and if so, how does WEFix handle such situations?

2. The paper mentions that WEFix is generally efficient, but acknowledges the potential for performance overhead in certain cases due to the dynamic adjustment of the wait window. Could you provide more details on the scenarios where this overhead is most likely to occur? How does WEFix mitigate this issue to maintain overall efficiency in web testing?

**Ethics Review Description:**

/

**Reviewer Confidence:**

2: The reviewer is willing to defend the evaluation, but it is likely that the reviewer did not understand parts of the paper

**Scope:**

4: The work is relevant to the Web and to the track, and is of broad interest to the community

---

### Official Review · Reviewer_fBZd · 2023-11-25

**Novelty:** 5
**Technical Quality:** 5

**Review:**

Summary
The authors design and develop an approach named WEFix to automatically fix the UI-based flakiness in web e2e testing. Specifically, WEFix leverages browser UI changes to predict the client-side code execution and generate proper wait oracles. By applying WEFix to 122 web e2e flaky tests from real-world web projects, the results show that WEFix can effectively find and fix flaky tests.

Strengths
- The authors propose a new approach to fix flaky web e2e tests.
- The authors implement a prototype of the proposed approach named WEFix.
- The effectiveness and performance of WEFix are evaluated on real-world web e2e tests. The tool does find several flaky tests and can fix most of these flaky tests.
- The artifacts of WEFix are open-source.

Weaknesses
- The confirmation on the fixed flaky tests has not been disclosed.

Detailed comments
I am quite positive about this paper. The approach proposed and the prototype implemented by the authors can effectively fix flaky web e2e tests. I just have a few suggestions and comments on the current version of the paper.

1. Evaluating and discussing the effectiveness of Algorithm 1.
Since some of the values used in Algorithm 1 are set heuristically, the author should evaluate and prove these values are set properly.

2. Disclosing the uncovered flaky tests and including the feedback in the paper.
The authors should follow the responsible disclosure policy to disclose their findings to the corresponding parties, such as developers and maintainers of web projects under evaluation. If the corresponding parties can help confirm the correctness of detection results, the authors could add the feedback in the paper to further show the effectiveness of WEFix.

**Questions:**

N/A

**Reviewer Confidence:**

3: The reviewer is confident but not certain that the evaluation is correct

**Scope:**

3: The work is somewhat relevant to the Web and to the track, and is of narrow interest to a sub-community

---

### Decision · Program_Chairs · 2024-01-22

**Decision:**

Accept

**Comment:**

All the reviewers are quite positive about the paper, considering well-organized structure and clarity. The reviewers also highlight the real-world findings and fixings of flakiness in web e2e testing. Yet, the reviewers do concern some minor issues, which, if fixed, could enhance the overall quality and clarity of the work.

 The authors are recommended to discuss the heuristic values set in the paper, which may significantly influence the performance. The authors may further clarify the evaluation settings, i.e., lack of comparison with other similar techniques, and limited evaluation on different browsers and platforms. Potential overhead/false positives/false negatives should also be discussed in the paper. The authors may also mention the feedback from developers about the reported flakiness.